# SCALABLE EVOLUTION STRATEGIES FOR IMPROVED HIERARCHICAL REINFORCEMENT LEARNING

## ABSTRACT

This paper investigates the performance of *Scalable Evolution Strategies* (S-ES) as a *Hierarchical Reinforcement Learning* (HRL) approach. S-ES, named for its excellent scalability across many processors, was popularised by OpenAI when they showed its performance to be comparable to the state-of-the-art policy gradient methods. However, to date, S-ES has not been tested in conjunction with HRL methods, which empower temporal abstraction thus allowing agents to tackle more challenging problems. In this work, we introduce a novel method that merges S-ES and HRL, which allows S-ES to be applied to difficult problems such as simultaneous robot locomotion and navigation. We show that S-ES needed no (methodological or hyperparameter) modifications for it to be used in a hierarchical context and that its indifference to delayed rewards leads to it having competitive performance with state-of-the-art gradient-based HRL methods. This leads to a novel HRL method that achieves state-of-the-art performance, and is also comparably simple and highly scalable.

## 1 INTRODUCTION

*Reinforcement learning* (RL) has been used to create artificially intelligent agents for tasks ranging from robot locomotion (Haarnoja et al., 2018) to video games such as StarCraft (Vinyals et al., 2019) and board games such as chess and Go (Silver et al., 2018). Many such agents use *Markov Decision Process* (MDP) based learning methods, such as Deep Q-Networks (DQNs) (Mnih et al., 2015) and the policy gradient family of methods (Sutton et al., 1999a; Sutton & Barto, 2018). MDP and gradient based RL methods are also used by hierarchical RL (HRL) algorithms, which are a class of RL algorithms that excel at challenging RL problems by decomposing them into sub tasks, which mimics the way we as humans build new skills on top of existing simpler skills. HRL methods have seen success in solving some of the hardest RL environments such as Montezumas revenge (Vezhnevets et al., 2017; Badia et al., 2020) and generating complex robot behaviours (Nachum et al., 2018). Another area of reinforcement learning that has enjoyed recent success is evolution strategies. These are a family of black box evolutionary optimization techniques, which Salimans et al. (2017) showed are competitive with non-hierarchical MDP based RL methods in the robot locomotion and Atari domain. The success of such approaches has lead to a wider use of ES for tackling RL problems, but it has yet to be used to solve challenging HRL problems.

ES has been used as a black box optimizer for a multitude of problems such as minimizing the drag of 3D bodies (Beyer & Schwefel, 2002), optimizing designs in structural and mechanical engineering problems (Datoussaïd et al., 2006), robot locomotion (Salimans et al., 2017; Conti et al., 2017; Katona et al., 2021) and loss function optimization (Gonzalez & Miikkulainen, 2020). There are many different flavours of ES (Beyer & Schwefel, 2002) each with different selection, mutation and self-adaption properties, for example CMA-ES (Hansen & Ostermeier, 2001) and $(1+\gamma)$-$ES$ (Beyer & Schwefel, 2002). However, in this work we are concerned with the version proposed by Salimans et al. (2017), namely *Scalable Evolution Strategies* (S-ES) because of it's proven performance in the domain of robot locomotion and Atari game playing. All flavours of ES follow the scheme of sample-and-evaluate, where it samples a *cloud* of policy variants around it's current policies parameters, evaluates these sampled policies to obtain a fitness and uses fitness to inform an update to the current policy. S-ES specifically uses fitness to approximate the gradient and moves the current policy parameters in the direction that maximizes the average reward. Given that ES is both a black-box process (making it indifferent to temporal details) and is a gradient free method it suffers from

sub optimal sample efficiency, however Liu et al. (2019) showed promising results addressing this inefficiency using trust regions which allow for more of a monotonic improvement.

S-ES obtained results comparable to MDP methods on a set of standard MuJoCo (Todorov et al., 2012) and Atari (Mnih et al., 2015) benchmarks (Salimans et al., 2017), however there are many RL problems harder than these standard benchmarks, some of which are near impossible to solve using non-hierarchical RL (in this paper we refer to this as flat RL) and others that are unsolved using flat RL. These environments can range from games such as Montezuma's revenge (Mnih et al., 2015) which is challenging because it requires long-term credit assignment, to robot locomotion, navigation and interaction (Nachum et al., 2018; Florensa et al., 2017) which requires complex multi-level reasoning.

HRL has long held the promise of solving much more complex tasks than flat RL methods. It allows policies to abstract away large amounts of complexity and focus on solving simpler sub goals. One of the first HRL methods is known as the options framework (Sutton et al., 1999b) which allows the controller policy to select the most appropriate primitive policy from a pool of primitive policies, this primitive passes control back to the controller once its actions are completed and the process repeats. Another competing HRL framework is feudal-RL (Dayan & Hinton, 1993), this framework allows for communication between the controller and primitives by having the controller set goals for the primitive to complete. Recent feudal-RL methods such as FeUdal Networks for HRL (FuN) (Vezhnevets et al., 2017) and HRL with Off-Policy Correction (HIRO) (Nachum et al., 2018) have shown a lot of promise for learning sparse reward problems and hierarchies requiring complex primitives. HIRO in particular takes the approach of using a two-level hierarchy (one controller and one primitive) where the controller sets the goal and reward for the primitive. For example the goal can take the form of a position an agent must reach and the reward is based on the agents distance to the goal position. HIRO, FuN and most modern HRL methods use MDP based RL methods to optimize their hierarchy of policies (Vezhnevets et al., 2017; Nachum et al., 2018; Sutton et al., 1999b; Badia et al., 2020) and to the best of the authors knowledge, non-MDP based RL solvers, such as ES, have not been extensively tested on hard RL problem that are typically reserved for MDP based HRL solvers.

ES has multiple advantages over MDP based RL methods, but two of these advantages make ES especially suited for HRL problems. First, it is invariant to delayed rewards and second, it has a more structured exploration mechanism (Salimans et al., 2017; Conti et al., 2017) relative to MDP based RL methods. Its robustness to delayed rewards is especially useful for HRL problems as much of the difficulty of these environments can come from the long term credit assignment problem. Similarly hard RL problems often have many large local minima, requiring intelligent exploration methods in order to be solved. These advantages suggest that ES and specifically S-ES should perform well on challenging HRL problems, however to the best of the authors knowledge S-ES has not yet been applied to HRL problems.

In this paper we introduce a new method[1] for training a two-level hierarchy of policies which are optimized using S-ES[2], namely *Scalable Hierarchical Evolution Strategies* (SHES). It will be evaluated on two difficult environments, which require robust robot navigation, locomotion and planning. We compare our method's performance to other MDP based HRL methods that have been tested on the same environments (Nachum et al., 2018; Vezhnevets et al., 2017). This paper aims to demonstrate that SHES performs well on environments that are challenging for MDP based HRL methods and that S-ES is as viable as any MDP based RL method when training hierarchies of policies. The results obtained in this work show that SHES provides the HRL space with a high task performance, highly scalable and comparably simple HRL method. Furthermore, our method achieves these results using the same hyper-parameters as its flat RL counterpart.

## 2    SCALABLE HIERARCHICAL EVOLUTION STRATEGIES (SHES)

In this section we present our framework for learning hierarchical policies using S-ES, which we call SHES. We show how current MDP based HRL methods needed to be adapted in order to work with

---

[1]*Removed for review, code can be found in supplementary material (ScalableHrlEs.jl)*

[2]*Removed for review, code can be found in supplementary material (ScalableEs.jl)*

S-ES and explain the important design choices taken, such as choice of primitive reward function, the goal encoding and controller architecture.

## 2.1 POLICY HIERARCHY

SHES is a Feudal RL (Dayan & Hinton, 1993) style method where a higher level policy set goals for a lower level policy. Dayan & Hinton (1993) use a multilevel feudal hierarchy, but similarly to HIRO, SHES uses a two level hierarchy consisting of a higher level controller policy $\mu^c$ and a lower level primitive policy $\mu^p$. The controller is responsible for setting goals and cannot directly perform actions in the world while the primitive directly controls the agent by taking actions in the world and attempts to reach the goals set by the controller. More formally, given a state $s_t$ from the environment the controller produces a goal $g_t \in \mathbb{R}^d$ (where $d$ depends on the goal encoding, in the case of this work $d = 3$). The controller produces $g_t$ every $c$ steps, in the interim the goal is transformed using a static function such that it is always relative to the current state, for example if $g_t$ is a vector from the agent to a target position, the static function will make sure that $g_t$ is always relative to the agents current position. The controller interval $c$ is kept as a hyper-parameter since it has been observed that learning $c$ often leads to it degenerating into the simplest cases where $c$ becomes 1 or the maximum episode length (Vezhnevets et al., 2017).

This provides the controller with a level of temporal abstraction which, in the case of the environments tested in this work, allows the controller to plan a path without worrying about how the agent will follow this path. The primitive is passed the goal $g_t$ and the state $s_t$ and tasked with reaching the goal. It samples an action $a_t \sim \mu^p(s_t, g_t)$ from its policy which is applied to the agent. The controller receives a reward from the environment, however it is also responsible for rewarding the primitive. As discussed in section 2.2 the primitive is sensitive to this reward and thus it should be chosen carefully. In HIRO, FuN and this work, the primitive reward is based on its distance to its goal $g_t$ (Vezhnevets et al., 2017; Nachum et al., 2018), however all such previous work makes use of different rewards.

In a strict feudal RL setting, rewards are not shared between controller and primitive. For example, if the primitive reaches the goal set by the controller, but this does not provide a high environmental reward then the primitive will receive a high reward, but the controller will not. Both this work and HIRO follow this strict style of primitive rewards. Even though FuN shares rewards between their primitives and controllers (Vezhnevets et al., 2017), this was decided against as it introduces a hyper-parameter to balance primitive and controller reward scales, which can be challenging.

Our method is similar to HIRO, though SHES differs in its lack of off-policy correction, it's primitive goal encoding and the use of S-ES to train the primitive and controller. This has the effect of making SHES very simple to implement, the only difference between SHES and S-ES being that there are now two policies that co-evolve instead of a single policy. SHES stores a set of parameters for both the controller $\theta^c$ and primitive $\theta^p$. Every generation it creates $n$ new pairs of controllers and primitives by perturbing the parameters $\theta^c$ and $\theta^p$. The perturbation is done by adding a small amount of noise sampled from an n-variate Gaussian to the parameters $\theta_i^c = \theta^c + \epsilon^c \sim \mathcal{N}(0, \sigma^2)$. The primitive is perturbed similarly using a different noise vector which is sampled from the same Gaussian $\epsilon^p \sim \mathcal{N}(0, \sigma^2)$ allowing for the sharing of common random numbers at no extra memory cost when compared to a single policy S-ES as SHES uses a shared noise table, which is one of the main contributions of S-ES (Salimans et al., 2017).

The sharing of common random numbers was shown by Salimans et al. (2017) to allow for near linear speedup when scaling up to 1000 CPU cores, which means that SHES should scale as well as S-ES, however it was only tested up to 240 cores[3]. Each pair of perturbed controller and primitive are evaluated in the environment such that the controller is given a fitness equal to the cumulative environmental reward and the primitive is given a fitness of its cumulative reward from the controller. Both primitives and controllers are separately ranked and shaped according to their fitness using the same method as Salimans et al. (2017) for S-ES. This is used to approximate the gradients for the controller and primitive separately using the ADAM optimizer (Kingma & Ba, 2014) and update the the controller ($\theta^c$) and primitive ($\theta^p$) parameters which is no different to how S-ES updates its policy.

---

[3]Each node has a Intel Xeon 24 core CPU at 2.6GHz, 120GB of RAM and nodes are connected by FDR InfiniBand

Using this type of Feudal RL method the controller poses goals for the primitive who's behaviour is constantly changing. This amounts to a non-stationary problem for the controller since while the controller is trying to learn the behaviour of the primitive its behaviour is changing, which can be challenging. Nachum et al. (2018) develop an off-policy correction method to combat the non-stationary problem and to allow for off-policy training and thus better sample efficiency. We found that we did not need a special method to combat this problem as S-ES robustness to noise makes this problem simpler to solve, since the primitives changing behaviour can be interpreted as noise by the controller, however this does come at the cost of sample efficiency.

---

**Algorithm 1:** SHES

---

1 **Input:** Learning rate $\alpha$, noise standard deviation $\sigma$, rollouts $n$, initial policy parameters $\theta^c$ and $\theta^p$

2 **for** $t = 0,1,2...$ **do**

3     **for** $i = 1,2..n$ **do**

4        Sample $\epsilon_i^c, \epsilon_i^p \sim \mathcal{N}(0, I)$

5        $F_i^c, F_i^p = F(\theta_t^c + \epsilon_i^c * \sigma, \theta_t^p + \epsilon_i^p * \sigma)$

6     **end**

7     $\theta_{t+1}^c = \theta_t^c + \alpha \dfrac{1}{n\sigma} \sum_{i=1}^n F_i^c \epsilon_i^c$

8     $\theta_{t+1}^p = \theta_t^p + \alpha \dfrac{1}{n\sigma} \sum_{i=1}^n F_i^p \epsilon_i^p$

9 **end**

---

## 2.2 PRIMITIVE REWARD

The manner in which the primitive is rewarded can have a large impact on the overall performance of SHES, where at the most basic level the reward needs to incentivise the agent to reach a target. In the literature there are many different ways that the primitive reward has been formulated (Vezhnevets et al., 2017; Nachum et al., 2018; Coumans et al., 2013), from this and our own informal experiments we've found the main components of a well performing primitive reward are incentive to reach the target consistently and quickly while avoiding local minima.

HIRO uses the most simple primitive reward, by rewarding the primitive with its negative distance to the goal $g_t$ (Nachum et al., 2018), this encourages the agent to move to the target quickly, however it introduces a challenging local minima where the agent can simply die instantly thus avoiding accumulating anymore negative reward. FuN rewards its primitive based on the cosine similarity of the path the agent has taken since the goal was suggested and the straight line from the agent's position to the goal (Vezhnevets et al., 2017). This encourages the agent to be very consistent which makes it more predictable for the controller, but this reward puts little emphasis on speed. Another possible reward which is used in both pyBullet (Coumans et al., 2013) and MuJoCo (Todorov et al., 2012) locomotion environments is the agent's velocity towards the target, while this does encourage fast movement this often comes at the cost of consistent paths, making it difficult for the controller to recommend positions. Rewarding the agent based on the percent of the total distance it covered (since the position was recommended) plus a bonus for reaching the target was found to be the best performing primitive reward and is what SHES uses.

$$R_t^p = 1 - d_t/d_c + (1 \; if \; d_t < L \; else \; 0)$$

where $d_t$ is the euclidean distance between the agent and the goal $g_t$ at timestep $t$, $d_c$ is the distance at timestep $c$ (the most recent timestep at which the controller recommended a goal) and $L$ is a distance threshold ($L = 1$ in the case of this work). This improves upon simply rewarding the primitive with the negative distance by: allowing it to be positive if the primitive performs well thus avoiding local minima and normalizing the distance thus making it agnostic to target distance. Also, adding an extra reward for being close to the target incentivises the agent to reach the goal as quickly as possible in order to maximize the amount of time it receives this extra reward.

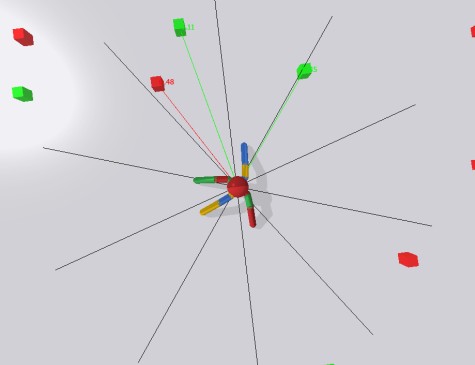

Figure 1: Sensor example in the *Ant Gather* environment. Two black lines enclose an area correlated with a position in the input vector. The closest object of interest between the black lines is given an intensity based on its distance to the agent and is added to its input vector position. If no objects are between two lines or within sensor range the value of the input vector at the related position is 0.

### 2.3 THE GOAL

SHES manages the primitive goal similarly to HIRO in that a new goal is recommended once every $c$ steps by the controller and for the next $c - 1$ steps this goal is transformed using a fixed goal transition function. In SHES the goal is the vector from the agent's position to the goal, thus the fixed transition function simply updates this goal each step given the agent's new position. However, the SHES goal differs from the HIRO goal in that it only recommends an $x$ and $y$ position in space, whereas in HIRO the goal passed to the agent is the entire state space, such that the primitive must attempt to match the position of all of the agents joints as well as the overall position of the agent. This approach is very general, but it limits types of primitive rewards that are usable and it is thus often more challenging for the primitive to learn this representation.

Similar to primitive rewards we found goal encoding to have a large impact on performance. The most obvious goal encoding to use would be a vector from the agent's position to the goal $g_t$. We found that this did not work since the values are not normalized, thus the primitive ANN performs worse because of non-normalized input data (Sola & Sevilla, 1997). However, normalizing the goal vector comes with its own issue since the agent no longer has any notion of distance to the goal $g_t$. We solve this by passing to the primitive, the distance to the target which is scaled down to an appropriate range (by dividing it by 1000).

This simple normalized vector goal encoding can be improved upon, taking inspiration from pyBullet's directional encoding (Coumans et al., 2013) we encode the primitive goal as the $sin$ and $cos$ of the angle from the agent to the goal $g_t$. This was done by allowing the controller to output a relative vector from the agent to the target and transforming this vector into an angle from the agent to the target, the $sin$ and $cos$ of this angle is the goal $g_t$ that is passed to the primitive. Similar to the normalized vector encoding this also provides the primitive with no notion of distance to the target and as such we pass the scaled distance to the target along with this goal encoding. This angle encoding can be seen as a normalization step, while also encoding more intuitive information about the direction to the goal.

### 2.4 ONE HOT CONTROLLER

Using a *one-hot* inspired encoding for the output of the controller helps simplify many problem-spaces via matching the output of the ANN controller to part of its input. To explain the one-hot inspired encoding consider the output of the controller network ($\mu^c$) is a vector $o \in \mathcal{R}^d$ such that each element of $o$ corresponds to one of $d$ equally spaced angles around the agent, this can be seen as the black lines in figure 1. In the same style of one-hot encoding popularly used as the output of image classifying convolutional neural networks (Potdar et al., 2017) the element with the highest value $o_h$ is selected and used to determine the point that the controller will recommend. To use figure 1 as an example $d$ would be 10 and each element in the output vector $o$ would correspond

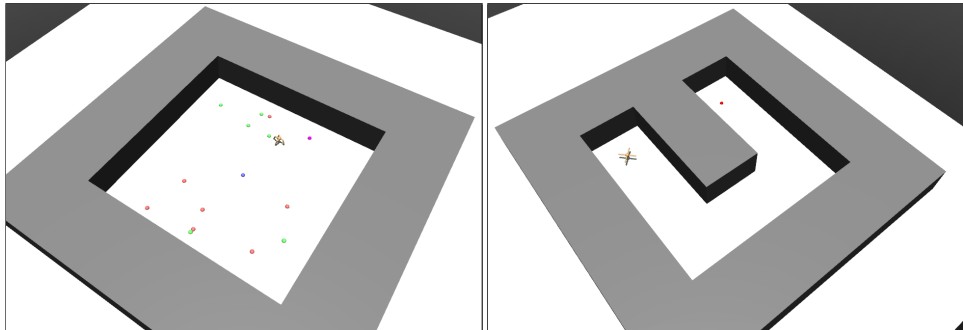

Figure 2: *Ant Gather* environment (left) and *Ant Maze* environment (right). The two environments used to evaluate the performance of SHES. In the gather environment the agent must learn to collect as much green *food* as possible while avoiding the red *poison* (the pink dot indicates the primitive goal $g_t$ and the blue dot is the agents starting position). In the maze environment the agent must learn to reach the red dot on the other side of the maze.

to one of the black lines. If each line is $k$ units long, given the highest output $o_h \in [-1, 1]$ the controller recommends the point $o_h * k$ units along the relevant line, where $k$ is a hyper-parameter representing the maximum distance from the agent that the controller can recommend.

This isn't a vital part of SHES, but it is especially relevant to the environments tested and can be used in agent-environment sensory interactions. This ANN architecture choice only works well for environments with sensors, since the input to the ANN is similar to its output. For example, if the agent observed a vector representing absolute positions of objects (instead of their sensor readings) this style of ANN architecture would not perform well. Only one of the two environments tested uses agent sensors and as such the one hot encoding is only used for this environment. When one-hot encoding is not used the controller simply outputs a vector representing the relative $(x, y)$ position from the agent's current position that the agent should move towards.

## 3 EXPERIMENTS

SHES was evaluated on two different environments that have sparse rewards and require both robot locomotion and navigation. The first environment is called *Ant Gather* and the second *Ant Maze*, an example of both can be seen in figure 2.

**Ant Gather:** The agent is given a positive reward for each green *food* it collects and a negative reward for each red *poison* it collects. Food and poison are generated in random positions each time the environment is initialized.

**Ant Maze:** The agent must learn to move to the opposite side of the maze. During training the agent must reach a randomly generated target inside the maze and is rewarded based on the negative distance to this target. During testing the agent must come sufficiently close to the red dot on the ultimate step of the episode to receive a score of 1 otherwise it gets 0.

To compare SHES we use results from Nachum et al. (2018), since this paper collected results on many different HRL methods such as HIRO (Nachum et al., 2018), FuN (Vezhnevets et al., 2017), *Stochastic Neural Networks* for HRL (SNN4HRL) (Florensa et al., 2017) and *Variational Information Maximizing Exploration* (VIME) (Houthooft et al., 2016). Results from these environments were taken after the agents had trained for 10 million steps. The environments used by Nachum et al. (2018) were originally written in python, but for this work have been faithfully recreated (using the same assets and physics engine) in Julia[4] for computational speed-up purposes.

|  | **Ant Gather** | **Ant Maze** |
|---|---|---|
| SHES (100) | $0.79 \pm 0.34$ | $0.0 \pm 0.0$ |
| SHES (600) | $2.88 \pm 0.61$ | $0.24 \pm 0.10$ |
| SHES one-hot (100) | $1.48 \pm 0.34$ | n/a |
| SHES one-hot (600) | $\mathbf{3.68 \pm 0.60}$ | n/a |
| HIRO (10) | $3.04 \pm 1.49$ | $\mathbf{0.99 \pm 0.01}$ |
| FuN cos similarity (10) | $0.85 \pm 1.17$ | $0.16 \pm 0.33$ |
| SNN4HRL (10) | $1.93 \pm 0.52$ | $0.0 \pm 0.0$ |
| VIME (10) | $1.42 \pm 0.90$ | $0.0 \pm 0.0$ |

Table 1: Task performance of comparative methods on *Ant Gather* and *Ant Maze*. Performance is the average of 10 randomly seeded trials with standard error. Values in parentheses are millions of training steps. One-hot encoding was not used for agents-environments without sensors (section 2.4). Performance of all methods that are not SHES are taken from Nachum et al. (2018).

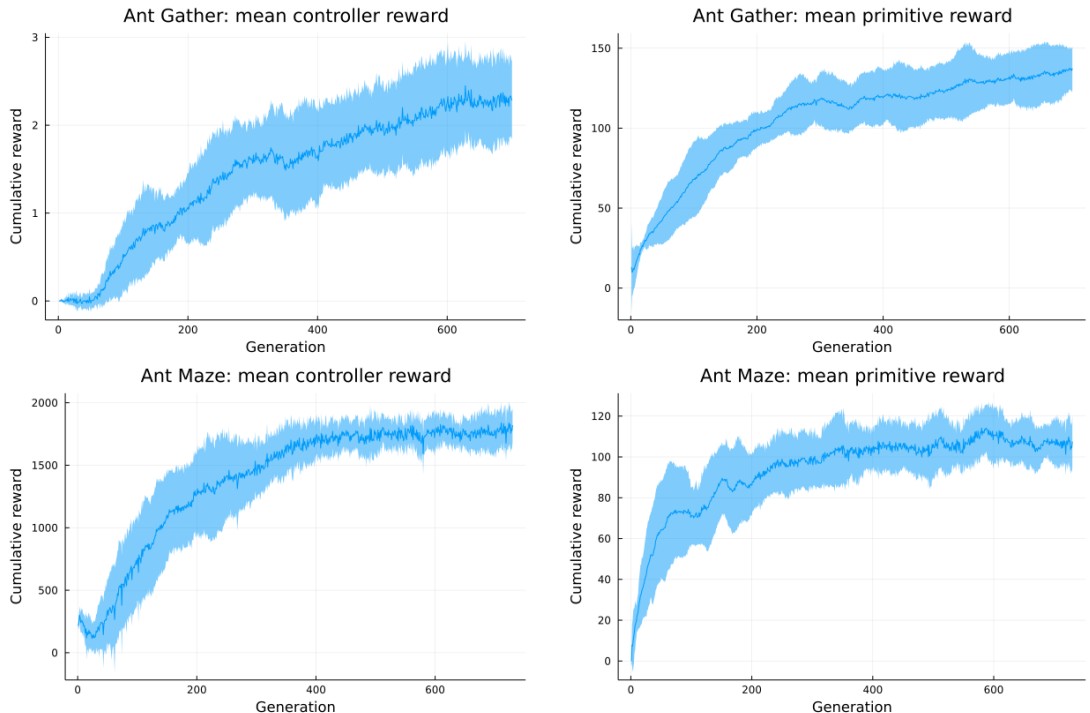

Figure 3: Cumulative training reward per generation over 10 SHES runs (mean of the mean). Shaded area shows standard error. X-axis is shown in generations, not in total steps.

## 4 DISCUSSION

Results in table 4 indicate that SHES performs suitably well, however the difference in sample efficiency is also quite clear. At 600 million steps SHES needs 60 times more samples than gradient based methods. The lack of sample efficiency is not unexpected as gradient free optimization is often less efficient than gradient based methods (Sigaud & Stulp, 2019). Specifically, the original S-ES paper (Salimans et al., 2017) saw sample efficiency up to 7.88 times worse than TRPO on simple 2D locomotion environments. Thus, S-ES was already less sample efficient than an on-policy gradient based method, meaning that the sample efficiency difference would be more exaggerated when compared to an off-policy algorithm such as HIRO. The 7.88 times more samples required were measured on simple 2D locomotion environments, when moving to the more challenging 3D Ant Gather and Maze environments this only amplifies the sample efficiency difference. Nonetheless

---

[4]*Removed for review, code can be found in supplementary material (HrlMuJoCoEnvs.jl)*

it doesn't mean one should not use SHES or S-ES. In simulated learning applications, task performance, learning speed, learnt behavioral robustness and the simplicity of SHES more than makes up for its sample inefficiency.

One of the main ways that SHES makes up for it's lack of sample efficiency is it's overall task performance, having a state-of-the-art mean score on Ant Gather of 3.68 when using a one-hot encoding and without a one-hot encoding being within one standard deviation of HIRO (the previous state-of-the-art) (Nachum et al., 2018). This clearly shows the benefit of the one-hot encoding, allowing for an easier representation for the controller to learn and improves upon the already good results from the non-one-hot version of SHES. Also, task performance on Ant Maze still achieves a respectable score given the difficulty of the problem. Results analysis indicates that learning starts quite late in the process, having a mean evaluation score of 0 at 100 million steps and 0.24 at 600 million steps. This is likely because during training, target positions are placed randomly and thus the agent does not get many samples of targets close to the evaluation goal after it has learned to walk around the corner. Along with its state-of-the-art performance, even after 600 million steps, SHES is still able to continue learning. This is supported by figure 4, showing a clear trend upwards for both the primitive and controller on Ant Gather. Also, results in table 4 indicate that for SHES applied to Ant Maze, learning also starts late, thus indicating that SHES has the capacity to continue improving late into the learning process. This is in contrast to HIRO which learns and plateaus rapidly (Nachum et al., 2018).

Despite the fact that SHES requires many samples, for all tests, 600 million steps never took longer than 12 hours on a 24 core CPU and given the simplicity of SHES it is very easy to parallelize its computation across multiple CPUs in order to obtain even more speed-up. The ease of parallelization along with the fact that S-ES needed no hyper-parameter or methodological changes in order to fit into SHES shows its robustness for all types of RL tasks and frameworks. Another area where SHES is exceedingly robust is with respect to the non-stationary learning problem that the controller faces. As can be seen in figure 4 for both environments the controller has very few dips in task performance meaning that it easily deals with the ever changing nature of the primitive. Thus, overall, results demonstrate that SHES is a robust HRL method which is able to achieve state-of-the-art performance on HRL tasks, while maintaining excellent scalability, and being comparably simple to implement, and needing no hyper-parameter modifications from its flat RL counterpart.

## 5 CONCLUSION

We have presented a novel method (SHES) for using S-ES in a two-level hierarchy which allows it to solve complex RL tasks requiring both robot locomotion and navigation. SHES achieved competitive performance with state-of-the-art methods in all environments tested. SHES needed no hyper-parameter tuning and demonstrated fast and scalable performance, providing a viable gradient free learning method for hierarchies of policies on challenging RL tasks. To the best of the authors' knowledge, this is the first time a hierarchy of ES has been used to solve such challenging RL tasks. Future work will include combining SHES with work done by Liu et al. (2019) to improve sample efficiency and testing it on more environments encapsulating a broader range of RL task types.

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

## A  APPENDIX

### A.1  ENVIRONMENT DETAILS

Environments were implemented using Julia's wrapper for MuJoCo, Lyceum (Summers et al., 2020). The implementation parameters were taken from the HIRO paper appendix (Nachum et al., 2018) and from their repository[5]. Similarly to HIRO we set the simulator to have a $dt$ of 0.02 and frameskip of 5.

#### A.1.1  ANT GATHER

This environment uses a modified version of the most modern ant robot to match the one used by HIRO. In addition is allowed to observe all *qpos* and *qvel*, the current timestep and the extra sensor observations so that it can observe the food and bombs.

Episodes are terminated when the ant falls or at 500 steps.

The reward is simply the number of apples collected minus the number of bombs collected.

#### A.1.2  ANT MAZE

The same slightly modified ant is used for this environment. It observes all *qpos* and *qvel* the current timestep and the target position $T_x, T_y$. The agent is placed in a $U$-shaped maze and each episode must reach randomly generated targets inside the maze. This is done slightly differently to the HIRO implementation, as in that implementation positions are sampled that could potentially be unreachable by the agent, in this implementation all sampled positions are reachable. During evaluation the agent must reach a specified position on the opposite side of the maze. The reward each steps corresponds to the negative l2 distance from the agent to the target.

For this environment the episode only terminates at 500 steps.

---

[5]https://github.com/tensorflow/models/tree/master/research/efficient-hrl

## A.2 IMPLEMENTATION DETAILS

### A.2.1 NETWORK STRUCTURE

The same network structure as S-ES is used, which is a simple feed forward neural network with one hidden layer of 256x256 units and $tanh$ activation. When the one-hot encoding is not used the outputs of the controller are scaled by the *controller distance* hyper-parameter.

### A.2.2 HYPER-PARAMETERS

Note that for all parameters relevant to S-ESs original implementation we use the same values, namely noise table size, $\sigma$, action noise standard deviation, learning rate and l2 coefficient.

- episodes per policy: 5
- policies per generation: 256
- controller interval: 25
- controller distance: 5 (the max distance from the agents position that the controller can recommend a goal)
- noise table size: 250000000
- $\sigma$ (standard deviation of Gaussian policy perturbing noise): 0.02
- action noise standard deviation: 0.01
- learning rate: 0.01
- l2 coefficient: 0.005

### A.2.3 IMPACT OF CONTROLLER INTERVAL

The controller interval $c$ or the frequency with which the controller recommends positions to the primitive is a sensitive parameter and can have a large impact on the performance of the overall system. A high interval (200) does not give the controller a lot of control, but it does allow the primitive to learn easily, in contrast a low interval (5) gives the controller fine grained control, but is too difficult a task for the primitive to learn. In practice a balance must be struck between these two extremes where the controller has enough control to effect the outcome and the primitive is able to learn to reach the goals set by the controller. We found 25-50 to be an ideal range for $c$, however Nachum et al. (2018) used a $c = 10$ which did not work for SHES. Thus the setting of this hyperparameter cannot be generalised to any feudal RL system and our recommended range only pertains to SHES.

