# OpenReview forum: "Evolution Strategies as an Alternate Learning method for Hierarchical Reinforcement Learning"
_ICLR.cc/2022/Conference — ICLR 2022 Submitted_

### Official Review · Reviewer_up7h · 2021-10-31

**Correctness:** 3
**Technical Novelty And Significance:** 1
**Empirical Novelty And Significance:** 2
**Recommendation:** 3
**Confidence:** 4

**Main Review:**

This paper tackles an important research direction of reinforcement learning. While the paper is clearly written, I have a few concerns and suggestions.

1) This paper misses an important prior work [Jain et al. 2019]. This prior work also applies evolution strategy (Augmented Random Search) to hierarchical learning. The prior work required less prior knowledge: the interface between the high and the low levels is learned, and the update frequency between the high and the low levels is also learned automatically. In this paper, both are specified manually. In addition, the prior work is validated on a real robot while this paper is tested in simulation. To improve the quality of this paper, a thorough discussion and quantitative comparison with [Jain et al. 2019] is needed.

2) The result does not significantly outperforms the state-of-the-art (HIRO). From Table I, In Ant Gather, the proposed algorithm achieve slightly better result but uses 60 times more samples. In Ant Maze, HIRO's performance is far better, even with only 1/60 of samples needed. The paper claims that ES is more scalable. To support this claim, a direct comparison of training time (wall-clock) between HIRO and ES should be provided.

3) The paper calls policy gradient methods as MDP methods multiple times in the Introduction. MDP is a problem formulation, and is not a solution. This paper is also using MDP formulation, but applies a derivative-free optimization solver. I believe what the paper intends to say is "policy gradient methods" that uses back-propgation to calculate gradient.

4) For Figure 3, it would be nice to plot learning curves of all baselines in same figures for easy comparison. If the curves of baselines are too short due to sample efficiency, it is OK to plot the curves with wall-clock time as the x-axis since an argument in this paper is that ES algorithms are more parallelizable and thus take less wall-clock time to train.

Reference:
Jain et al., Hierarchical Reinforcement Learning for Quadruped Locomotion, 2019

**Summary Of The Paper:**

This paper proposes to apply evolution strategy to hierarchical reinforcement learning. The high-level controller sets goals and the low-level primitive learns to move to the goal. The high-level and the low-level have different reward functions. The paper applies evolution strategy to solve such a problem. The proposed method is evaluated on two tasks: Ant Maze and Ant Gather. It outperforms the baselines in one of the experiments.

**Summary Of The Review:**

This paper proposes a straightforward application of evolution strategy on hierarchical learning. The novelty is limited. It misses an important prior work [Hierarchical Reinforcement Learning for Quadruped Locomotion, Jain et al. 2019], which also uses ES (ARS) on hierarchical learning for locomotion. The prior work seems to achieve better results than this paper. In addition, comparing to HIRO, it is hard to conclude that the proposed algorithm outperforms the state-of-the-art.

---

### Official Review · Reviewer_BgoU · 2021-11-02

**Correctness:** 2
**Technical Novelty And Significance:** 2
**Empirical Novelty And Significance:** 2
**Recommendation:** 3
**Confidence:** 4

**Main Review:**

Strengthes:
* Simple search strategy for hierarchical policies allowing highly parallel implementation

Weaknesses:
* Quality of the evaluation of the proposed approach is below the standard of the community. (1) The comparison in Table 1 is rather unfair. The authors compared the performances of variants of the proposed approach after 100 or 600 million steps, with the performances of existing approaches after 10 million steps. (2) The two tested environments are similar, and do not support the generality of the proposed approach. It is also unclear whether these environments are representative for the environment said to be "hard" by the authors repeatedly in the introduction. (3) Ablation studies are missing.
* Novelty is questionable. The search strategy is almost the same as [Salimans et al, 2017]. The only difference is that the fitnesses for the controller policy and for the primitive policy are different. The fitness of the primitive reward is rather similar to existing approaches except for a way to normalize it. The contribution of this proposed primitive reward is not evaluated in experiments, hence unclear.
* Clarity of the statements should be improved. (1) F in Algorithm 1 is not explained in detail. (2) The authors repeatedly says "challenging problems" and "hard problems", but it is not clearly stated what kinds of difficult tasks the authors are targeting. (3) The authors claims that ES is invariant to delayed reward. But it is not clear what is meant by it. (4) The authors say that "hard RL problems often have many large local minima" and ES is advantageous for this perspective. This might be true for the standard ES, but for the one used in this paper, the authors set the noise std for the parameter search to be 0.02, meaning that this approach do not take into account so much global information of the objective function.

**Summary Of The Paper:**

This paper proposes a novel approach for hierarchical reinforcement learning. As the title says, the authors claims that the use of evolution strategies is useful to train hierarchical policy. Basically, the search is based on the estimation of the gradients of the controller fitness and the primitive fitness using the gaussian samples around the current parameters. It is mostly the same as [Salimans et al, 2017]. The authors also introduce the primitive reward. The performance of the proposed approach is evaluated on two test environments.

**Summary Of The Review:**

Based on the above review, the novelty and the significance are questionable. The statements should be improved for clarity. The experimentation should be improved to support the claim of the paper.

---

### Official Review · Reviewer_xRrU · 2021-11-02

**Correctness:** 3
**Technical Novelty And Significance:** 3
**Empirical Novelty And Significance:** 2
**Recommendation:** 5
**Confidence:** 4

**Main Review:**

This work presents the novel algorithm which applies ES to the HRL scheme. The ES part shares key concepts and features with the previous work, and the hierarchy part uses a two-level hierarchy similar to the previous works.

Strengths
- The combination of ES and HRL is novel and interesting
- The proposed algorithm is straightforward and seems easy to implement
- Experimental results are comparable to the previous algorithms

Weaknesses
- The core idea consists of two main parts, similar to previous works, S-ES and HRL.
- Experiments are only conducted in two environments, some results are not compatible with previous algorithms.

This paper is well-organized and easy to follow. Their claims are reasonable but seem not fully supported by the experiments. Also, as the authors pointed out, the algorithm needs much more samples to achieve similar results. Future work will enhance the sample efficiency and test in more environments.

One of the main contributions of this paper is to show the effectiveness of ES in HRL problems. The authors provide the experimental results but I think it is not enough to show the effectiveness. They present the results in two environments, "Ant Gather" and "Ant Maze". In "Ant Gather", the algorithm shows comparable results with previous works, although it requires 10x to 60x training steps. However in "Ant Maze", the result of "SHES one-hot" is not available, and "SHES" shows low performance. Also, the closest previous work, HIRO, presents two more environments, but those environments are not shown in this work.

**Summary Of The Paper:**

The paper presents the integration of evolution strategies with hierarchical reinforcement learning. From the natural characteristics of ES algorithms, it also can be highly scalable, so authors named the algorithm as SHES (Scalable Hierarchical Evolution Strategies). The algorithm is tested in two robot locomotion and navigation tasks.


**Summary Of The Review:**

Although the proposed methods are novel, it seems that the experimental results need to be improved.

---

### Official Review · Reviewer_zZsS · 2021-11-02

**Correctness:** 3
**Technical Novelty And Significance:** 2
**Empirical Novelty And Significance:** Not applicable
**Recommendation:** 5
**Confidence:** 4

**Main Review:**

Strengths:
- As the authors highlight, ES is invariant to the timescale of rewards. This is indeed a useful property in HRL, and one that should be exploited further.
- As per the previous point, the paper was an interesting reading.
- Performance on the tested environments is promising.
- <informal on> I am a big fan of ES. :) <informal off>

Weaknesses:
- Evaluation is limited to just two environments. In particular, it would have been interesting to see a comparison of the various methods on traditional (pre-deep) HRL environments, and in general on a larger breadth of tasks.
- Overall, the degree of novelty of the work presented seems limited, as the method presented is basically a standard Feudal RL setup optimized via ES instead of gradient-based methods.
- Comparison of the final performance of the proposed method vs the competing methods seems a bit biased, as the competing methods were only trained for 10 million steps, while the method from the authors was trained for 600 million steps.
   While the problem of sample efficiency of ES is a valid point, it is not clear whether the competing methods would have achieved comparable or higher performance if they were trained on the same amount of data.




**Summary Of The Paper:**

The authors propose a method for Hierarchical Reinforcement Learning based on the OpenAI variant of Evolution Strategies.
The proposed method follow the HRL approach of Feudal RL, implementing two controllers at different levels of abstraction: a master controller outputs goals (every number of steps) that the slave controller learns to achieve The two controllers receive different rewards to incentivize each to learn its task.


**Summary Of The Review:**

The main issue with the paper is that the degree of novelty seems limited, as it seems that in practice the proposed method is just Feudal RL optimized via ES instead of gradient-based methods. Also evaluation is insufficient.

---

### Decision · Program_Chairs · 2022-01-20

**Decision:**

Reject

**Comment:**

This paper presents the use of scalable evolution strategies (S-ES) in hierarchical reinforcement learning. After reviewing the paper and reading the comments from the reviewers, here are my comments:

- The proposal is quite novel. It requires major improvements to clearly state how this proposal contributes in the field.
- The main concern is about the experimental results. There are some flaws in the comparative results. Also, they do not support the proposal.